# Effect of Minimum Milking Interval on Traffic and Milk Production of Cows Milked by a Pasture Based Automatic Milking System

**DOI:** 10.3390/ani12101281

**Published:** 2022-05-17

**Authors:** Françoise Lessire, Jean-Luc Hornick, Isabelle Dufrasne

**Affiliations:** Fundamental and Applied Research on Animal and Health, Animal Production Department, Faculty of Veterinary Medicine, University of Liège, Quartier Vallée 2—Avenue de Cureghem, 6, 4000 Liège, Belgium; jlhornick@uliege.be (J.-L.H.); isabelle.dufrasne@uliege.be (I.D.)

**Keywords:** automatic milking system, grazing, pasture-based, milking permission, traffic to the robot

## Abstract

**Simple Summary:**

Several studies have demonstrated that combining grazing and robotic milking is possible. However, there is often a decrease in milking frequency, which leads to a decrease in milk production. The objective of this study was to investigate the effect of two strategies to improve traffic in a pasture-based automatic milking system. Therefore we formed four groups differing based on concentrate allocation and based on minimum milking interval (MMI) necessary to access the robot for milking. Therefore four groups (high concentrate–short MMI, high concentrate–long MMI, low concentrate–short MMI, low concentrate–long MMI) were constituted. We compared these four groups with regard to traffic parameters (milkings, refused milkings) and animal production. The study highlighted the positive effect of high concentrate–short MMI on traffic to the robot by reducing the number of refused milkings. High concentrate allocation allowed to maintain high milk production over the experiment duration.

**Abstract:**

In dairy farms automatic milking systems and grazing, traffic to the robot is the cornerstone of profitability as higher milking frequency enhances milk yield. In this study, we investigated whether shortening the minimum milking interval (MMI), i.e., the required time between two milkings for an animal to get access to the milking unit, coupled with high concentrate allocation, could increase the daily milking frequency (MF, milking/cow/day) and consequently the milk yield of grazing cows. Two groups of cows (*n* = 19 and *n* = 20) belonging to the same herd were discriminated based on concentrate supply (high vs. low: 4 vs. 2 kg/cow/day) and then further divided on the basis of MMI (4 h vs. 6 h) so that four groups were formed (HC4 h–HC6 h–LC4 h and finally LC6 h). Higher concentrate allocation induced a rise in milk yield (MY, kg/cow/day) and allowed to stabilize it in periods of grass shortage but did not influence milking frequency, while shorter MMI (4 h) was correlated with higher MF without effect on MY. A combination of both strategies (4 h and high concentrate) improved the traffic globally to the robot. This result was linked to a reduction of refused milking and, therefore, the decrease in returns to the robot. This strategy could be advised to maximize the system’s efficiency during periods of high milk sales. When the economic conditions do not favour the increase in concentrate supply, short MMI could facilitate the traffic and increase the efficiency of returns.

## 1. Introduction

The dairy sector is facing numerous challenges worldwide, including increasing herd size, high milk productivity coupled with volatile sale prices and automation of some practices, e.g., automatic milking systems (AMS) [1,2]. In parallel, the consumers pay more attention to animal welfare and the sector’s environmental impact. In this context, grazing demonstrates several advantages, including lower milk production costs [3,4], some environmental benefits [5] and a positive image toward consumers [6,7]. Coupling AMS and grazing is possible, and several ways to achieve it were described in the literature [8,9,10,11,12]. Yet it remains challenging. Indeed, the profitability of AMS is linked to enhanced milk yield related to higher milking frequency (MF). In AMS, cows are milked on a voluntary basis. The average MF rises to 2.6 to 2.7 milkings/cow/day, which is 0.6 to 0.7 milkings more than in a conventional milking system. However, at grazing, MF is reported to drop around 2 milkings/day, so milk production increases to a lesser extent and even decreases [9,10,13]. Reaching higher MF seems crucial. Moreover, cows in pasture-based robots tend to demonstrate more marked gregarious behaviour than in the barn. Optimising the traffic to the robot is thus a key point. Several parameters were reported to influence traffic to the robot-like concentrate supply, pasture allocation and minimum milking interval (MMI). This last parameter is defined as the minimum time elapsed between 2 consecutive milkings to access the AMS. Providing enlarged access time, i.e., low MMI, to the robot is expected to increase MF. This study aimed to determine whether a high concentrate supply coupled with enlarged access time to the robot allowed to maintain MF and milk yield (MY) at a similar level to that observed at the barn.

## 2. Materials and Methods

### 2.1. Animals and Experimental Design

The study was conducted at the Experimental Farm of Sart Tilman, University of Liège, Belgium (5.58° E, 50.42° N) from 9 May 2015 till 30 June 2015 (53 days). During the study periods, the experimental herd totaled an average of 52 cows (Min: 40; Max: 56) milked on pasture by a mobile AMS Lely A3 as described by Lessire et al., 2017 [14].

The cows’ diet was composed of grazed grass and a variable amount of concentrate (Moulins Bodson, Villers l’Evêque, Belgium) provided in the AMS during milking. This feedstuff was composed of 18.5% whey pellets, 11.5% dried beet pulps, 4% spelt, 10% barley, 24.5% wheat, 5% wheat distillers, 4% beet molasses and 4% soybean meal. It provided per kg dry matter (DM) 170 g crude protein (CP), 242 g starch + sugars and 894 VEM (1475 kcal net energy lactation). 

The cows grazed in a single herd. For this study, the herd was randomly divided into 3 groups of 10 and 1 group of 9 cows, taking into consideration the lactation number (LN), the days in milk (DIM) and the average MY (kg/cow/day) recorded at one week before the setting up of the trial, to get a maximum of similarity. We verified that there were no statistically significant differences between the groups. In a first step, two groups were designed based on high and low levels of concentrate allocation, i.e., HC (high concentrate allocation), 4 kg/cow/day and LC (low concentrate allocation), 2 kg/cow/day, respectively. The distribution was foreseen based on DIM and MY with a progressive increase from 1 kg to 2 kg and to 4 kg/cow/day in LC and HC, respectively, from DIM = 1 to DIM = 70 d. At DIM > 70 d, only MY determined the concentrate supply with a minimum of 0 and 3.5 kg/cow/day at MY ≤ 18 kg in LC and HC, respectively. 

Thereafter, different MMI were attributed to programming the robot computer. HC and LC groups were separated into sub-groups based on MMI, each HC and LC being allocated to 2 groups with different MMI, i.e., short MMI (MMI 4) = 4 h and long MMI (MMI 6) = 6 h. A total of 4 groups (HC4 h, HC6 h, LC4 h and LC6 h) were thus obtained, whose characteristics are demonstrated in Table 1. Thirteen cows were not included in the study for different reasons (low MY, high refusal rate, …) and received low concentrate allocation with an MMI of 6 h.

### 2.2. Grazing Management

The cows had 24 ha of permanent grasslands at their disposal divided into 15 paddocks from 0.6 to 3.1 ha (Figure 1). Grazed grass was composed mainly of perennial ryegrass (*Lolium perenne*) and white clover (*Trifolium repens*). The herd grazed as a whole, without physical separation between the groups. The meadows were divided into 2 blocks, with day (A) and night (B) allocation (AB-design). Selection gates at the exit of AMS were changed at 7 am and 4 pm and directed the animals from the night pasture block (NP) to the day one (DP) and conversely. Access to water was provided in day/night pasture blocks with respect to animal welfare. Water access was also provided near the milking platform.

In each DP and NP, a strip was moved daily to allow access to fresh grass to achieve an estimated grass consumption target of 17 kg DM/cow/day. 

Nutritional quality of grass was checked by random hand-sampling on the grazed pastures. The samples were oven-dried (65 °C for 72 h) and analysed by NIRS for composition prediction (CP, NDF, ADF, lignin, water-soluble carbohydrates (WSC)) in order to determine the nutritional value according to the Dutch feeding system as described by De Boever [15]. 

Animal performances (MY, kg/cow/visit, amount of consumed concentrates (kg/cow/visit) and the data relative to the traffic to the AMS as number of milkings per day or successful milking (SM/day), number of failed milkings (FM/day) if robot failed to attach milking cluster) and number of refused milkings (RM/day) occurring if the delay between two visits is insufficient were recorded by the transponders fixed on an HR-tag neck collar (SCR, Netanya, Israel) and identifying each cow. Milk yield and amount of consumed concentrate were calculated by adding milk produced and concentrate amount at every milking over a 24-h period (0000 to 2400 h). Robot visitations were calculated by adding SM, FM, and RM. Milking interval was defined as the time between 2 milkings, i.e., visits where milk was collected. 

The grass intake was estimated post hoc using the procedure described hereafter: Sward height was measured by an electronic rising plate meter (Jenquip^®^, Feilding, New Zealand) before (1) and after the cows (2) had accessed the paddock. Herbage density (3) was estimated by mowing a strip of 0.38 m × 10 m and collecting the mowed sward, drying at 65 °C for 72 h and weighing it. Grass height (4) was measured on this strip with the rising plate mater. The collected sample was expressed in kg DM/ha and then divided by the grass height value: (3) divided by (4) so that the sward density was expressed in kg DM/cm/ha (5). The sward density ((5), kg DM/cm/ha) was multiplied by the average grass height value (1 and 2) to estimate the grass stocks at entry and exit (6 and 7). The grass stocks were finally divided by the number of cows and length of stay (8 and 9). The difference between the in and out (9)–(8) represented thus the consumed grass in kgDM/cow/day.

Weather data were collected at a station close to the experimental site (5 km) and included T°, relative humidity (RH, %) and rainfalls (mm). The data were recorded every 6 h. During periods of high T °C, the temperature-humidity index (THI) was calculated following the formula applied by Bernabucci et al., 2014 [16] to assess the risk of heat stress (HS). Mild heat stress was considered at THI values >69 described as impacting animal welfare [16,17]

### 2.3. Statistical Analysis

The statistical analyses were performed using SAS 9.3 (SAS Institute Inc., Cary, NC, USA). The data were analysed according to the PROC MIXED procedure using a repeated statement = week of observation on random factor = animal and covariance analysis type compound symmetry (cs).

The following model was applied
Yijkl = μ + Gr_i_ + week_j_ + LN_k_ + ClasDIM_l_ + Gr_i_ × week_j_ + e_ijkl._
where µ = mean, Gr = group effect. This parameter was in a first analysis divided into 4 classes (i = 1, 2, 3, 4: HC4 h, LC4 h, HC6 h, and LC4 h, week_j_ = week of measurement (j from 1 to 8). In a second step, the effect of concentrate allocation (HC vs. LC) and of milking permission (MMI4 vs. MMI6) were studied by merging subgroups (e.g., HC4 and HC6 vs. LC4 and LC6). 

LN: lactation number (k = 1 to 3 with 1 = primiparous, 2 = 2^d^ lactation and 3 more than 2 lactations. ClasDIM: stage of lactation (l = 1: DIM < 150; l = 2: DIM from 150 to 250 and l = 3 for DIM > 250.

The interactions between week and Gr were analysed, and e_ijkl_ represents the residual error~(N [0,σ_²_).

Yijkl was tested for MY, MF, number of failures and refused milkings, and milk collected by milking and milking interval.

The statistical significance level was set at *p* < 0.05, *p*-value comprised between *p* > 0.05 and <0.10 were considered as trend.

We used the Proc Freq procedure to draw contingencies tables for χ^2^ tests. Each table performed included information about the count of observed value, estimation of expected value, row percentage and column percentage. The calculation of adjusted residuals was chosen to highlight the most interesting results, according to the procedure described by Sharpe et al., 2015 [18].

## 3. Results

### 3.1. Grazing Management

The mean sward height at the entrance and exit was 12.2 ± 2.7 cm and 8.0 ± 1.7 cm. The average grass growth was 43.4 ± 14.1 kg DM/ha/day (Max: 61.8 on 13 May 2015—min: 21.4 kg DM/ha/day on 30 June 2015). The average grass stocks were estimated at 4032 ± 1023 kg DM/ha. Weather conditions were unusual: The study period included 37 days without rain, i.e., more than half of the study period. A period of 22 consecutive days without rain was recorded from 26 May 2015 to 17 June 2015 (Figure 2). On 22 June 2015, the most important rainfall was registered (32 mm, W7), representing half of the total rainfalls observed during the study (70 mm, from W1 to W7). After this day of rain, we noticed a return to the drought that lasted until the end of the study. The mean T °C was 14.7 °C (min: 9.2 °C, 16 May 2015—Max: 23.6 °C, 5 June 2015). These weather conditions influenced the grass stocks, as demonstrated in Figure 3.

The average nutritional values of the grazed grass are presented in Table 2. The CP value was lower than observed before in the same site (142 g/kg DM vs. 168 g/kg DM and 160 g/kg DM in 2013 and 2014, respectively), while WSC content was higher (211 g/kg DM vs. 175 and 187 g/kg DM in 2013 and 2014, respectively). The other parameters were within the usual values.

### 3.2. Zootechnical Performance

The statistical analysis of the MY did not demonstrate a group effect. On the other hand, the week and week × group effects were significant (*p* < 0.001 and *p* < 0.01, respectively). From W1 to W8, a loss of 3.36 kg of milk was observed, but this figure differed following the groups, as shown in Figure 3 (*p* < 0.01). The distinction between the groups according to the amount of concentrates distributed was very clear, while the duration of milking permission did not seem to have an influence. From W1 to W8, the HC groups (including MMI4 and MMI6) produced 2.60 kg less, while LC groups (including MMI4 and MMI6) recorded a production loss of 4.10 kg. The average difference between HC and LC in production due to concentrate allocation was on average 2.96 kg/cow/day (*p* < 0.01), while it reached only 0.18 kg/cow/day when MMI4 was changed into MMI6 (ns).

In accordance with these observations, we distinguished two groups based on concentrate allocation, i.e., HC (HC4 + HC6) and LC (LC4 + LC6), and performed a new statistical analysis. The results of this comparison are presented in Table 3. The difference in MY according to concentrate allowance was 2.93 kg milk/cow/day (*p* < 0.05). The difference in concentrate allocation reached 2.34 kg/cow/day (*p* < 0.001). The milk response (gain in MY/kg concentrate supplied) was 1.25 kg milk/kg concentrate. Regarding MY, milk per milking and concentrate allocation, no significant difference could be observed in short MMI (HC4 + LC4) vs. long MMI (HC6 + LCI6 (Table 4).

### 3.3. Voluntary Cows Traffic to the Robot

The MF was little affected by either MMI or the distribution of concentrates (2.26 vs. 2.13 milking/cow/day (−4%) in MMI4 and MMI6 respectively, *p* < 0.05 and 2.17 vs. 2.22 milking/cow/day in LC and HC (+2%) respectively; *p* < 0.10). The other indicators of robot traffic were not significantly impacted by concentrate supply. On the contrary, RM occurred less frequently (0.93 ± 0.04—20% less; *p* < 0.001) in MMI4 in comparison with 1.23 ± 0.03 in MMI6. This difference in RM influenced the visitations leading to a significant difference following the length of MMI (Table 4). A deeper analysis was performed to evaluate if one of the four groups formed at the beginning (HC4, HC6, LC4 or LC6) could be differentiated from the others by its trafficking behaviour toward the robot (Table 5). Refused milkings were almost doubled in HC6 in comparison with HC4, so visitations were the most important in this group. Thus, the short MMI allowed animals to restrict their movements to the robot. On the other hand, the LC6 group demonstrated the lowest MF and MY compared to other groups (Table 5). No difference in milking interval was noted, whatever the studied factor. The range of values for this parameter extended from 0.68 to 34.60 h, although these extreme values were marginal. Yet, the percentiles 5 and 95% are 4.82 h and 17.10 h.

### 3.4. Analysis of Contingency Tables

A first contingency table was constructed analysing the occurrence of milking and refused milkings according to the week of observation and group allocation. The comparison of SM and RM occurrence over the observation weeks is presented in Figure 4. The RM/SM ratio was estimated at 48% on average, indicating that one in three visitations leads to RM. Deviations from this average were observed in W1 and W7 with a higher proportion of RM (57 and 56%, respectively). In W7, this was coupled with an increased number of SM. In W8, on the other hand, the average percentage of RM fell to less than 25%. The following figures show the relative contribution of each group to the total of refused milkings in occurrence counts (Figure 5).

The contingency tables (SM × group × week) and (RM × group × week) were analysed to highlight the significant differences between expected and observed values. The confidence interval was set at 2.58 for a significance level of *p* < 0.01, following the recommendations of Sharpe et al., 2015 [18] for a contingency table including a large number of cells. None of the adjusted residuals for SM reached this value, signaling no significant difference between the observed and expected occurrence values. For RM, in 5 cells out of 32, the adjusted residual reached this significance level. In W2, the group HC6 showed a higher occurrence of refused milkings, while the contrary was observed for LC4. The group HC4 showed a huge drop in refused milkings in W5 and a significant increase in W7. In LC6, the only significant value was observed in W8 with a drop in refused milkings (Figure 5).

Concerning the repartition of passages to the robot, we have noted that 50% of the SM occurred at the times of change from NP to DP barriers and vice versa (8–12 h) and (16–20 h). Only 4% of SM happened in the early morning (0–4 h) (Figure 6). Passage frequency reached around 12% without statistically significant differences for time periods 4–8 h, 12–16 h, and 20–24 h. Few differences were observed in the period of SM between the different groups. Conversely, the majority of RM occurred between 12 h and 20 h (64%) with little discrepancies between the groups (less RM for HCI6 compensated by higher RM in LC4 between 20–24 h (Figure 7). The proportion of refused milkings to milkings was extremely high from 16–24 h (891 refused milkings for 1113 milkings, 80%) while it was only 22% from 8–12 h (257 refused milkings for 1211 milkings). The traffic to the robot is thus extremely tight at change from DP to NP. A contingency table was built to investigate whether the week influenced the time schedule of SM and RM. Week W1 was characterised by an exceptionally low number of SM (7.74%, i.e., 346/4468 SM in total). The affluence to the robot described in the time period 16–20 h extended to the 20–24 h time period for a total of 20% SM, and 31% RM. Week W7 counted an exceptionally high number of SM compared to the other weeks (almost 33% more). This increase was balanced throughout the day, except for a large influx in periods 4–8 h and 8–12 h representing 43% of the total SM, while the passages from 20–24 h are abnormally low (8.5%). Around 20% of the total recorded RM occurred during W7, with a higher proportion from 16–20 h (56%) and a lower proportion from 12–16 h (16%), the sum of these two figures represented 72% of the RM recorded during this week. The W8 showed a global passage time schedule different from the other weeks, especially with lower SM in the period 12–16 h (6%) compensated by higher passages from 20–24 h (21%). During W8, the number of RM was significantly lower (7.6%), with 50% of them happening from 16–20 h. The distribution of RM throughout the other weeks did not present high divergencies. Weather observations showed high T° during this week (from 23.8 °C on 25 June 2015 to 28 °C on 30 June 2015) combined with a mean HR of 66%. The THI values observed ranged between 74 to 81 from 12 to 18 h.

### 3.5. Effect of Parity

The MY was not influenced by the lactation number (LN1, primiparous; LN2, two lactations; LN3, more than two lactations). Conversely, all the parameters estimating the efficiency of traffic to the robot were influenced by this factor. The MF was higher in cows of more than two lactations (LN3, MF = 2.37 ± 0.02 milkings/cow/day vs. MF = 2.07 ± 0.03 and 2.17 ± 0.04 milkings/cow/day for LN1 and LN2 respectively; *p <* 0.001). The refused milkings occurred more frequently in multiparous (LN3: refused milkings = 1.32 ± 0.03/cow/day vs. refused milkings = 1.11 ± 0.04 and 1.09 ± 0.06/cow/day for LN:1 and 2 respectively). The number of returns increased following the lactation number from 3.02 ± 0.02 for LN1 to 3.30 ± 0.03 for LN2 and to 3.74 ± 0.02 for LN3. A contingency table was built to highlight the time schedule of SM and RM following the number of lactations. These tables helped highlight differences between expected and observed frequencies of events regarding LN. As shown in Figure 6, most of the milkings occurred in periods 8–12 h and 16–20 h. At these times, LN1 represented 44% (significantly more than expected) and 30.5% of the total milkings (significantly less than expected). From 20–24 h, LN1 milkings were the most frequent (54.7%—significant). Regarding LN3, milkings occurred most often in 4–8 h (68%—*p* < 0.05) and in 12–16 h and 16–20 h (about 50% of milkings—ns). Regarding LN2, no significant difference appeared between expected and observed occurrences. Few refused milkings were noted for LN1 between 4–8 h and 8–12 h (7.9 and 21% respectively—*p* < 0.05). Most LN1 refused milkings occurred in 16–20 h (50%—*p* < 0.05). LN3 refused milkings were over-represented in 8–12 h (64.2%—*p* < 0.05). LN2 showed no specific difference.

## 4. Discussion

This study aimed to determine whether high concentrate supply coupled with short MMI improved the traffic to the robot, i.e., increased MF and thus MY of grazing cows milked on pasture by AMS. The different factors that were likely to influence circulation to the robot, like weather conditions, grass depletion or lactation number, were also highlighted during the course of the study.

**Grass height measurement and calculation of available grass stocks** showed that the grass consumption target of 17 kg DM was met at the beginning of the grazing season, but grass stocks decreased during the study due to the drought. The nutritional content of the grass was comparable to usual values in terms of energy [11,19,20], although CP was low in comparison with other European publications [11,19,20]. The increased supply in concentrate led to higher MY in HC groups and limited the production loss over the study period. The milk response was higher than that reported by Delaby et al., 2001 [19] but similar to those reported by McKay et al., 2019 [21]. Higher milk response is frequently reported in link with low grass allocation [2,3]. This could also be influenced by the concentrate composition composed of more than 60% of cereals [21].

**Changes in MMI had no effect on MY**. The analysis of the results based on the sole criteria of concentrate distribution and MMI could be summarised as follows: concentrate supply affected MY but not the traffic to the robot, while the change in MMI impacted circulation to the robot but not the zootechnical performance. These results seem to contradict the concept of the highest MF, the highest MY [13,22,23]. However, deeper analysis of the results group by group (HC4, HC6, LC4 and LC6) suggests that combining the two strategies (HC—short MMI) is susceptible to improving traffic and zootechnical performance. In our study conditions, higher concentrate combined with MMI set at 4 h did not rise MF but reduced refused milkings so that the total returns to the robot were less frequent. At first glance, the MF seemed to be decreased by long MMI. However, a more thorough analysis showed that this outcome is mainly due to the results of LC6. The combination of long MMI and low concentrate induced a decrease in MF but stable refused milkings, which is contradictory to the results of HC groups (HC6 demonstrated the highest rate of refused milkings).

**The MMI4 h is shorter than in the other publications** evaluating MMI impact and comparing MMI6 vs. MMI12 [10,13,24]. To our knowledge, such a short MMI was never tested in a pasture-based robot before [9]. In practice, MMI is usually set at a minimum of 6 h [20,25,26] or only set at 4 h for early calved cows or in relationship with high MY [10,27]. Yet, short MI is reported to increase the risk of intramammary infection due to over-stimulation of teat sphincters [28]. The MMI set at 4 h could theoretically allow six milkings per cow and per day vs. four milkings per cow per day for MMI6. However, this number of milkings was not reached whatever the MMI. On the contrary, MMI4 h reduced the number of refused milkings and positively affected cow traffic. Consequently, less time was spent traveling to the robot, leaving more time for grazing and rumination. In our study, the average MI was estimated at 10.66 h (lsmeans), which is lower than in Lyons et al., 2013 [13] but in accordance with Nieman et al. 2015 [28] and Vandooren et al., 2004 [29]. The MI was not changed from 6 h to 4 h, confirming the limited milking frequency effect. Therefore a small risk of over-stimulation of the teats and limited impact on udder health could be expected. The distribution of concentrates did not influence MI. The range of variation of MI was less extended than in other studies [30].

**The distribution of the number of SM among the groups** showed that most of them occurred at the change of gates, confirming that the cows are mostly motivated to travel to the AMS by access to fresh grass, as described by other authors [10,23,24]. Low milking frequency observed from 0–4 h is usually reported in publications and is probably due to the diurnal behaviour of cows [31,32]. The gregarious behaviour of grazing cows could also affect the time schedule of attendance to the robot [33].

**The analysis of the week factor** showed that a larger number of SM and RM occurred during W7. During this week, prolonged dry conditions were interrupted by a day of heavy rain. Then the dryness resumed, accompanied by high temperatures (W8). The length of dry periods induced a marked decrease in grass growth, impacting grass stocks. Low grass availability was even more marked in W7 and W8. This observation is confirmed by the decrease in MY recorded at that time. We presume that the relative grass deficit led the cows to return more frequently to the robot. The influence of feed depletion on traffic was also described in other studies [10,25,34]. On the contrary, RM and SM occurred less frequently in W8. THI values of more than 68 were observed during this week, indicating mild HS. In addition, these stress periods coincided with the grazing of two meadows relatively far from the robot (DP: 14—NP: 2—Figure 1). The distance to be covered was approximately 1 km between both locations at the change of gates, compared to a distance of 650 m between the DP: 13 and NP: 6, the previous week. The influence of weather conditions on returns could be confirmed by the observed shift in the timing of the passages to the robot. Yet, SM occurred more frequently in 16–20 h instead of 12–16 h. Several authors investigated the influence of the distance to the robot with sometimes different conclusions. Some authors demonstrated that the distance between the grazed paddock and AMS did not influence MF [25,35]. However, in these studies, the longest examined distance from meadows to the robot was 850 m under favorable weather conditions. Another publication indicated an increase in MF and the returns to the robot during HS, but the distance between the meadows and the AMS was only about 180 m [36]. Conversely, other publications confirmed the influence of weather conditions on the time schedule of visits to a pasture-based robot [37]. The allocated paddocks during this period were very shaded (Figure 1). This could also induce a change in daily routine [38].

**The effect of parity was marked on traffic parameters with no change in MY.** Cows of more than two lactations circulated more frequently to the robot, with more than 3.5 returns/day to the robot. The ratio of refused milkings/SM was estimated at 57% compared to 43 and 45% for NL1 and NL2, respectively. According to our results, 50% of SM occurs at gate changes allowing access to fresh grass, i.e., from 8–12 h and 16–20 h. In our study, a large proportion (44%) of primiparous cows, i.e., usually considered low-ranked, accessed the robot between 8–12 h with a very low refusal rate. On the other hand, between 16–20 h, the proportion of primiparous cows was lower than expected (30%) but without impact on the refusal rate. At this time, cows with more than two lactations monopolised the robot, accounting for more than 50% of both milkings and refused milkings. The access to the robot was therefore clearly influenced by the hierarchy within the herd. Therefore, a higher proportion of primiparous cows were milked in the next period. These observations were in accordance with other studies that highlighted that low-ranked cows were prevented from milking during the highest traffic periods [39,40,41].

## 5. Conclusions

In this study, high concentrate allocation (from 2 to 4 kg/cow/day) increased the milk yield and made it possible to stabilise milk production during the length of the experimentation. The milk response was high compared to other studies, probably due to grass depletion. High concentrate supply induced no change in traffic parameters, but short milking intervals decreased refused milkings rate. In fact, the group combining a high concentrate supply and a short minimum milking interval (4 h) was the most efficient, with a ratio of refused/successful milkings of 38%. The time schedule of returns to the robot was influenced by the change of gates directing from night to day pasture block or, conversely, by the high affluence of cows at these times. This schedule was influenced by the parity with most frequent milkings for primiparous in 8–12 h time period while multiparous showed high circulation rate resulting either in successful or refused milkings from 16–20 h. During the length of the study, traffic to the robot appears to be influenced by climatic conditions, such as drought and consecutive grass depletion. The cows were probably more prone to return to the robot to receive the additional feed. During the last week of the study, lower traffic was noticed, and a change in the time of attendance at the robot. We hypothesized that it was due to drought and warmer temperatures inducing heat stress.

To conclude, shortening the minimum milking interval did not affect milk yield but improved the animals’ traffic flow. The relatively high concentrate allocation and short milking interval were beneficial regarding milk yield and traffic to the AMS. However, this result should be confirmed in studies conducted under other conditions, e.g., outside of drought periods.

## Figures and Tables

**Figure 1 animals-12-01281-f001:**
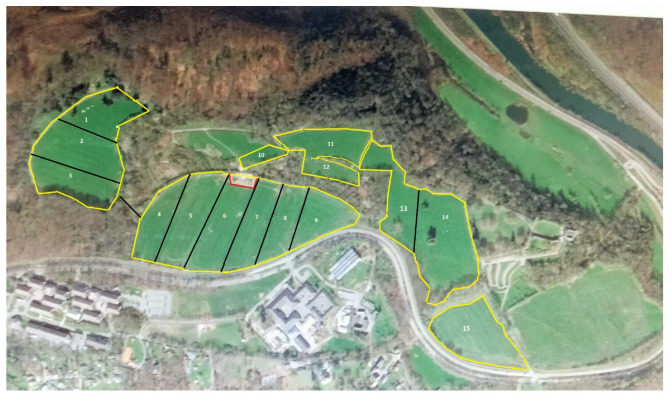
Paddocks layout. Twenty-four ha of pastures are divided into 15 paddocks ranging from 0.6 to 3.3 ha. Cows were allocated to the day (P7–P15) and night paddocks (P1–P6) by passing through a selection gate. In red; the position of the milking unit.

**Figure 2 animals-12-01281-f002:**
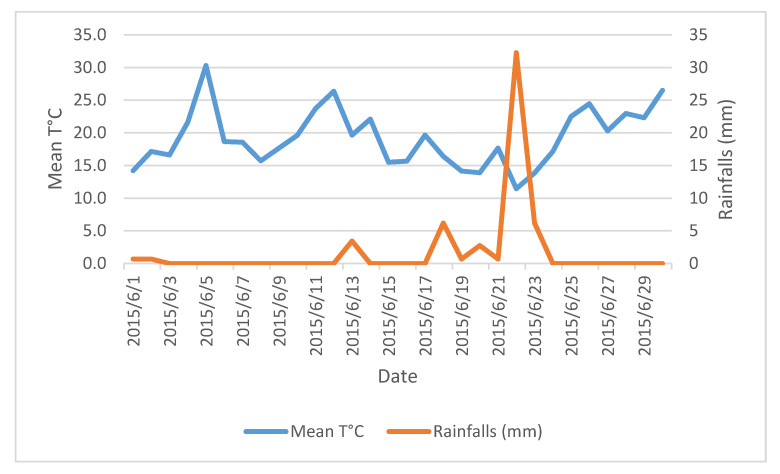
Weather conditions recorded in June 2015.

**Figure 3 animals-12-01281-f003:**
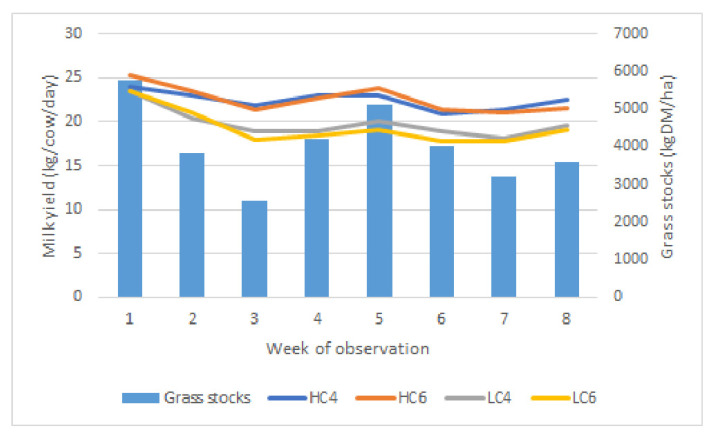
Evolution of milk yield over the observation weeks in the 4 groups in relation to the grass stock available (kgDM/ha) on the grazed paddock per week. Abbreviations. HC4, High concentrate allocation, minimum milking interval set at 4 h. HC6, High concentrate allocation, MMI set at 6 h. LC4, Low concentrate allocation, MMI set at 4 h. LC6, Low concentrate allocation, MMI set at 6 h.

**Figure 4 animals-12-01281-f004:**
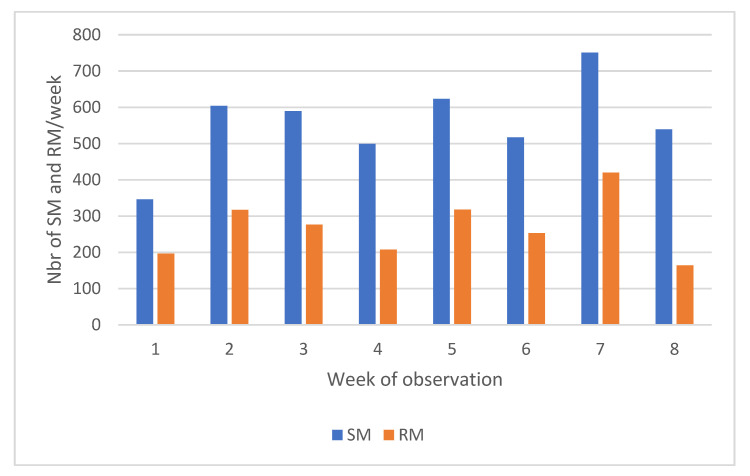
Relative occurrence of successful milkings (SM) and refused milkings (RM) over the weeks of observation.

**Figure 5 animals-12-01281-f005:**
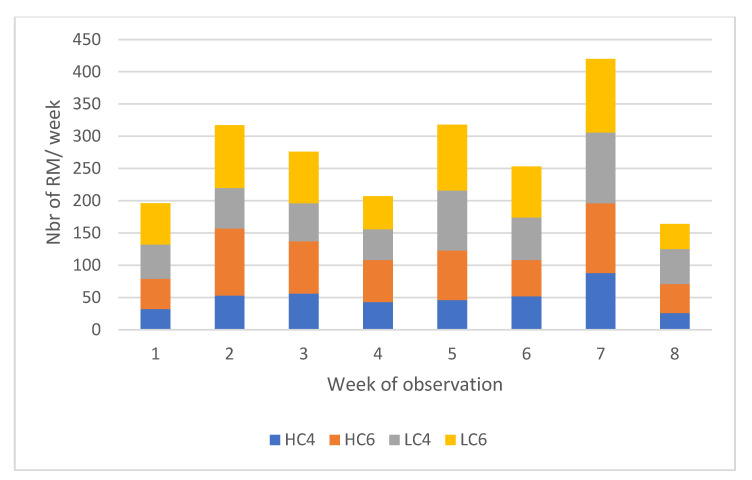
Relative occurrence of refused milkings (RM) of the different groups over the weeks of observation. Abbreviations. HC4, High concentrate allocation, MMI set at 4 h. HC6, High concentrate allocation, MMI set at 6 h. LC4, Low concentrate allocation, MMI set at 4 h. LC6, Low concentrate allocation, MMI set at 6 h.

**Figure 6 animals-12-01281-f006:**
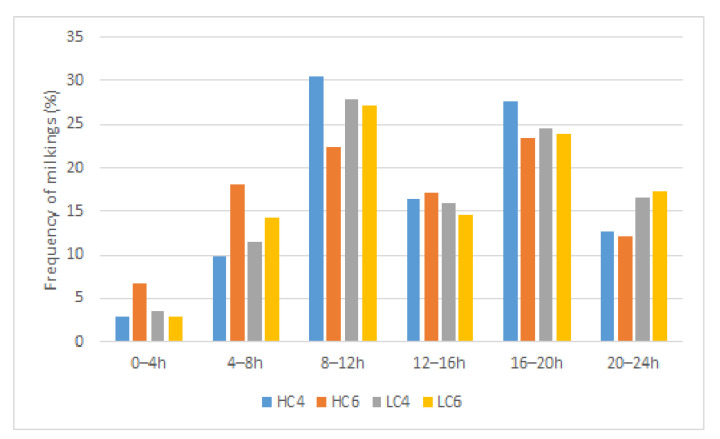
Time schedule of milkings (SM) of the different groups. Abbreviations. HC4, High concentrate allocation, MMI set at 4 h. HC6, High concentrate allocation, MMI set at 6 h. LC4, Low concentrate allocation, MMI set at 4 h. LC6, Low concentrate allocation, MMI set at 6 h.

**Figure 7 animals-12-01281-f007:**
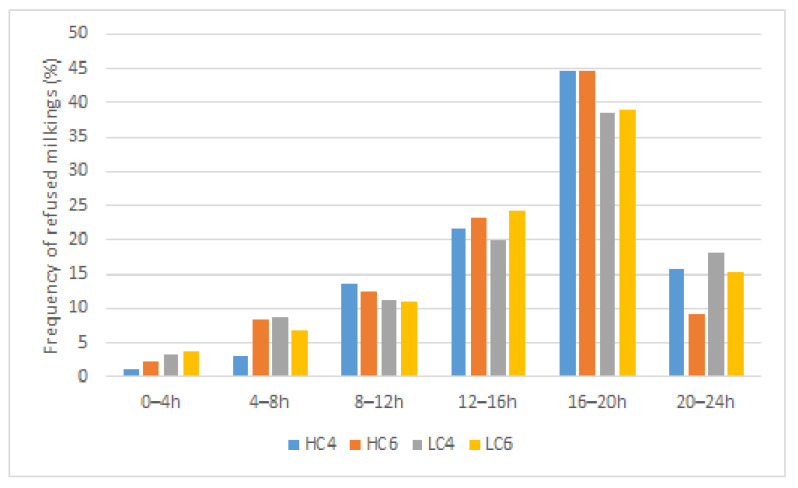
Time schedule of refused milkings (RM) of the different groups. Abbreviations. HC4, High concentrate allocation, MMI set at 4 h. HC6, High concentrate allocation, MMI set at 6 h. LC4, Low concentrate allocation, MMI set at 4 h. LC6, Low concentrate allocation, MMI set at 6 h.

**Table 1 animals-12-01281-t001:** Characteristics of the group at the beginning of the trial. Values are means ± standard deviation.

Groups	HC4	HC6	LC4	LC6
DIM (days)	135 ± 59	146 ± 114	144 ± 95	163 ± 52
LN	2.6 ± 1.4	2.4 ± 1.3	2.6 ± 1.7	1.9 ± 0.7
Primiparous (*n*)	3	3	4	3
MY (kg/cow/day)	26.9 ± 5.2	25.4 ± 7.6	25.6 ± 4,5	23.8 ±3.4
*n*	9	10	10	10

Abbreviations. HC4, High concentrate allocation, minimum milking interval (MMI) set at 4 h. HC6, High concentrate allocation, MMI set at 6 h. LC4, Low concentrate allocation, MMI set at 4 h. LC6, Low concentrate allocation, MMI set at 6 h. DIM, days in milk. LN, lactation number. MY, milk yield. Values determined at the beginning of the trial are means ± SD.

**Table 2 animals-12-01281-t002:** Nutritional grass values.

g/kgDM	DM	CP	NDF	ADF	Lignin	WSC	VEM
Mean	210	142	467	245	26	211	967
Std	38	18	43	28	3	32	48
*n*	15	15	15	15	15	15	15

Abbreviations. DM, dry matter. CP, crude protein. NDF, neutral detergent fiber. ADF, acid detergent fiber. WSC, water soluble carbohydrates. VEM, Voeder Eenheid voor. Melk, unit determining the net energy lactation of the feedstuff, i.e., 1000 VEM = 1650 kcal net energy lactation.

**Table 3 animals-12-01281-t003:** Comparison of zootechnical performance of dairy cows fed a low (2 kg/day) or high (4 kg/day) level of concentrate complementation.

Concentrate Complementation	Low	High	*p* > F
MY (kg/cow/day)	19.61 ± 0.94	22.54 ± 1.08	*
MY per milking (kg/cow/milking)	8.31 ± 0.35	9.68 ± 0.37	**
Concentrate (kg/cow/day)	1.39 ± 0.13	3.73 ± 0.14	***
SM (/cow/day)	2.17 ± 0.02	2.22 ± 0.03	trend
RM (/cow/day)	1.11 ± 0.04	1.04 ± 0.04	ns
FM (/cow/day)	0.02 ± 0.30	0.01 ± 0.20	ns
Visitations (/cow/day)	3.35 ± 0.02	3.33 ± 0.02	ns
Milking interval (h)	10.82 ± 0.33	10.15 ± 0.36	ns

Abbreviations. MY, milk yield. SM, successful milking. RM, refused milking. FM, failed milking. Values are lsmeans ± SE. *, *p* < 0.05; **, *p* < 0.01; ***, *p* < 0.001; Trend: *p* > 0.05 and <0.1.

**Table 4 animals-12-01281-t004:** Comparison of zootechnical performance according to the length of minimum milking interval (MMI).

Minimum Milking Interval	4 h	6 h	*p* > F
MY (kg/cow/day)	21.21 ± 0.94	20.94 ± 0.96	ns
MY per milking (kg/cow)	9.02 ± 0.36	8.98 ± 0.36	ns
Concentrate (kg/cow/day)	2.61 ± 0.13	2.51 ± 0.13	ns
SM (/cow/day)	2.26 ± 0.03	2.13 ± 0.02	*
RM (/cow/day)	0.93 ± 0.04	1.23 ± 0.03	***
FM (/cow/day)	0.01 ± 0.2	0.02 ± 0.2	ns
Visitations (/cow/day)	3.23 ± 0.02	3.45 ± 0.02	***
Milking interval (h)	10.78 ± 0.36	10.79 ± 0.33	ns

Abbreviations. MY, milk yield. SM, successful milking. RM, refused milking. FM, failed milking. Values are lsmeans ± SE. *, *p* < 0.05; ***, *p* < 0.001; ns, not significant.

**Table 5 animals-12-01281-t005:** Analysis of the trafficking behaviour of the four initially formed groups.

Groups	HC4	HC6	LC4	LC6
MF (milking/cow/day)	2.26 ± 0.03 ^a^	2.22 ± 0.03 ^a^	2.26 ± 0.03 ^a^	2.05 ± 0.03 ^b^
RM (/cow/day)	0.85 ± 0.05 ^a^	1.41 ± 0.04 ^b^	1.01 ± 0.05 ^c^	1.14 ± 0.04 ^c^
Visitations (/cow/day)	3.16 ± 0.03 ^a^	3.68 ± 0.02 ^b^	3.32 ± 0.03 ^a^	3.25 ± 0.02 ^a^
MY (kg/cow/day)	22.47 ± 1.33	22.57 ± 1.31	19.79 ± 1.26	19.33 ± 1.24

Abbreviations. HC4, high concentrate, MMI set at 4 h. HC6: high concentrate, MMI set at 6 h. LC4, low concentrate, MMI set at 4 h. LC6, low concentrate, MMI set at 6 h. MY, milk yield. RM, refused milking. Values are lsmeans ± SE. Different superscripts identify significant differences.

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
