# Peer review of "Effect of Minimum Milking Interval on Traffic and Milk Production of Cows Milked by a Pasture Based Automatic Milking System"

_animals, 2022, doi:10.3390/ani12101281_

Round 1
Reviewer 1 Report
Although grazing in dairy cows is considered fundamental, at least from the perspective of animal welfare, the priority is milk production, so the study of the combination between grazing and robotic milking is attractive, since robotics in dairy production is becoming increasingly important.
The drought that occurred during a large part of the experiment could bias the results and their conclusions. However, it should be noted that it is part of nature and it is good for authors to mark it.
It's an interesting line of research that we encourage to continue.
Author Response
Dear Reviewer,
I would like to thank you for your careful reading of this manuscript. I thank you also for the positive comments about our study and your encouragements.
Yours sincerely,
Françoise Lessire

Reviewer 2 Report
General comments
This is a manuscript regarding some factors (minimum milking interval, level of concentrate, etc.) affecting automatic milking systems performance under pasture conditions.
To my understanding the work is of interest, but the text seems too long, parts of the manuscript are not well written due to an inconsistent use of punctuation marks, where under the same situation authors put hyphen (-), comma (,) or semicolon (;), indifferently. For this reason, it is important to make a deep revision of the whole manuscript before publication.
In addition, I think that one of the key points is to check the abbreviations used to identify the 4 experimental groups, where authors use combinations of words difficult to remember (i.e. HCMMI4h) and they repeat the term MMI four times. In my opinion this could be simplified as: HC4h, HC6h, LC4h, LC6h.
If authors make these or similar changes, check all the manuscript, including Tables and Figures. They must be understable independently of the main text.
In addition, the term “refusals” is frequently used in this manuscript, what could make some confusion especially for readers working in other fields, as nutrition, where this word usually means non eaten food, or in this particular case it could mean non eaten concentrate. For this reason, when possible, would be of interest to reduce their use and say something like “milking refusals” or “refused milking” instead of refusals.
Finally, I have a concern. Authors say that 13 cows were not included in the study for different reasons (low MY, high milking refusal rate, …), and they received low concentrate allocation with a MMI of 6h. How can authors justify that these 13 cows using the same robot didn’t alter the behavior of the cows involved in this research?
Particular comments
Suggested text and/or corrections:
L 19: Put milking refusals instead of refusals
L 25: (MF, milkings/cow/day)
L 27: (high vs low: 4 vs 2 kg/cow/day)
L 28: 4 groups were formed (HC4h, HC6h, LC4h and LC6h).
This change in groups name should be done in the rest of the document (i.e. lines 89, 90, 165, 167, 168, 236, 381, etc.)
L 29: (MY, kg/cow/d)
L 31: shorter MMI (4h)
L 33: reduction of milking refusals
L 74-75 VEM (Voeder Eenheid voor Melk, 1,475 Kcal of net energy lactation?)
(indicate if VEM means metabolizable energy or net energy lactation)
L 81-82 Possible simplification (or even more): ....on basis of high and low level of concentrate allocation (i.e. HC, high concentrate, 4 kg/cow/d, and LC, low concentrate, 2 kg/cow/d) respectively.
L 82-84: Authors say that “distribution was foreseen on basis of DIM and MY with a progressive increase from 1 kg to 2 kg and to 4 kg/cow/d in LC and HC respectively from DIM = 1 to DIM= 70 d.”
I guess this was a pre-experimental period. In this case, would be of interest to indicate it, and clearly differenciate when started the true experimental period. I think in Table 1 appears that it was at 135 DIM, but I’m not sure of seeing it in the text.
In addition authors say (L 85-86) that “At DIM > 70 d, only MY determined the concentrate supply with a minimum of 0 and 3.5 kg/cow/d at MY≤ 18 kg in LC and HC respectively.”
Please, clarify if the level of MY was also taken in consideration during the experimental peridod as a factor to modify the concentrate fed to the cows of each group.
L 91-94: I would say: Thirteen cows were not included in the study for different reasons (low MY, high milking refusal rate, …) and they received low concentrate allocation with a MMI of 6h.
As indicate previously, please, try to justify that they didn’t affect the experimental work.
L 108: What do you mean with (AB-design) ?
L 126-130: Suggeted text: Animal performances (MY, kg/cow/visit), amount of consumed concentrates (kg/cow/visit) and the data relative to the traffic to the AMS as: number of milking per day or successful milking (SM/d), number of failed milkings (FM/d), if robot failed to attach milking cluster, and number of refused milkings (RM/d), occurring if the delay between two visits is insufficient, where recorded.....
L 132: ......were calculated by adding, respectively,......
L 145: ((5), kg DM/cm/ha)
L 192-195: Please, write “,” instead of “-", example:
rainfall was registered (32 mm, week 7), representing quite the half of the total rainfalls observed during the study (70 mm). After this day of rain, we noticed a return to drought 193 that lasted until the end of the study. The mean T°C was 14.7°C (min: 9.2°C, 16/5/2015 and max: 23.6°C, 5/6/2015).
L 202: Please, check if is necessary to define WSC.
L 208: Indicate if this kcal are of metabolizable energy or net energy lactation.
L 216-220: Indicate the P value if differences are significant (i.e. P < )....
L 232-234: Indicate the P value if statistical differences are significant (i.e. P < )....
L238: Remember that all Tables and Figures must be understable independently of the main text.
Suggested tittle:
Table 3. Comparison of zootechnical performance of dairy cows fed a low (2 kg/d) or high (4 kg/day) level of concentrate complementation
L 240-242: Check this part, there are different errors.
Suggested correction:
Abbreviations: MY, milk yield; SM, successful milking; RM, refused milking; FM, failed milking. Values are lsmeans ± SE. *, P <0.05; **, P <0.01; ***, P <0.001; Trend, P >0.05 and <0.1.
If you make this type of changes, do the same in the abbreviations of other Tables and Figures (L 246-247, etc…)
L 244: Table 4. Comparison of zootechnical performance according to the length of minimum milking interval (MMI).
L 275: …..according to the week of lactation….
L 283: I think it would be better to type: The contingency tables (SM x group x week) and (RM x group x week)
L 295 and 297: Type refused milkings (RM) instead of refusal
L 302-303: ….. (8-12h) and (16-0 h), respectively.
L 304: …. for time periods 4-8h, 12-16h, and 20-24h.
L 309: Suggested correction: (891 refusals for 1113 milkings, 80%)
L 329 and 334: Check titles of Figures 6 and 7. The week factor is not shown in these Figures, but it appears in the title.
L 342-345: Check the punctuation marks of this part.
L 353-358: When there are significant differences, indicate the P value (or P < ) instead of saying “significant”
L 368-369: Authors say that nutritional content of the gras was comparable to usual values, but would be necessary to be more specific, i.e. was comparable to usual values in terms of energy?
L 387-388: possible modification: .....but stable milking refusals, which is contradictory...
Author Response
Dear Reviewer,
First of all, I would like to thank you for your careful reading of this manuscript. I think the changes you suggest will be really helpful to improve the readability.
I made changes to punctuation marks to make them more consistent.
I changed the abbreviations used to identify the experimental groups as you suggest and I verified if the changes were made throughout all the text (HC4 instead of HCMMI4 , etc).
The details of the modifications are provided in the annexed document. For ease of reading I have already removed some of the modifications from the track changes.
I hope that these changes and explanations will meet your expectations.
Yours sincerely,
Françoise Lessire

Reviewer 3 Report
It has taken quite a while to get this data published but the results are still valid. The authors have carried out a well designed experiment and analyzed the data thoroughly.
My only negative remark concerns line 470 in the conclusion stating that the study did not cause alteration in udder health. This statement is not supported by observations or measurements and should be removed from the conclusions.
Author Response
Dear Reviewer,
I would like to thank you for your careful reading of this manuscript. The statement about udder health was removed from the conclusions accordingly to your suggestion.
Yours sincerely,
Françoise Lessire
